# Applicability of Pulsed Electric Field (PEF) Pre-Treatment for a Convective Two-Step Drying Process

**DOI:** 10.3390/foods9040512

**Published:** 2020-04-19

**Authors:** Robin Ostermeier, Oleksii Parniakov, Stefan Töpfl, Henry Jäger

**Affiliations:** 1Elea Vertriebs-und Vermarktungsgesellschaft mbH, Prof. von Klitzing Str. 9, 49610 Quakenbrück, Germany; 2Department of Food Science and Technology, University of Natural Resources and Life Sciences (BOKU) Vienna, Muthgasse 18, 1190 Vienna, Austria

**Keywords:** onion, convective drying, pulsed electric fields, quality, drying kinetics

## Abstract

Available literature and previous studies focus on the Pulsed Electric Field (PEF) parameters influencing the drying process of fruit and vegetable tissue. This study investigates the applicability of PEF pre-treatment considering the industrial-scale drying conditions of onions and related quality parameters of the final product. First, the influence of the PEF treatment (*W* = 4.0 kJ/kg, *E* = 1.07 kV/cm) on the convective drying was investigated for samples dried at constant temperatures (65, 75, and 85 °C) and drying profiles (85/55, 85/65, and 85/75 °C). These trials were performed along with the determination of the breakpoint to assure an industrial drying profile with varying temperatures. A reduction in drying time of 32% was achieved by applying PEF prior to drying at profile 85/65 °C (target moisture ≤7%). The effective water diffusion coefficient for the last drying section has been increased from 1.99 × 10^−10^ m^2^/s to 3.48 × 10^−10^ m^2^/s in the PEF-treated tissue. In case of the 85/65 °C drying profile, the PEF-treated sample showed the highest benefits in terms of process efficiency and quality compared to the untreated sample. A quality analysis was performed considering the colour, amount of blisters, pyruvic acid content, and the rehydration behavior comparing the untreated and PEF-treated sample. The PEF-treated sample showed practically no blisters and a 14.5% higher pyruvic acid content. Moreover, the rehydration coefficient was 47% higher when applying PEF prior to drying.

## 1. Introduction

The drying of onions is performed in order to reach a longer shelf life and to achieve a sufficient food safety level. Another purpose is to achieve lower storage and transportation costs by reducing volume and weight [1,2]. This is why drying processes are gaining increasing importance in the food industry [3,4]. According to Naderinezhad et al., 2016, the convective drying is the most applied thermal method for onion drying. However, the amount of energy and time required by drying processes within the food industry is significant [5,6,7]. The main influencing factors for energy consumption in terms of the drying process are the temperature, the time, as well as the initial moisture of raw material and the thermal efficiency of the drier [4,8,9]. One of the most difficult conflicts to solve in industrial drying is to achieve a preferably short drying time for economic and ecological reasons and simultaneously to preserve product quality even at high drying temperatures [4,9,10,11,12]. Onions are valued for their flavors and for their nutritional value, and these quality attributes should be maintained during processing [13,14]. Grewal et al., 2015 and Mujumdar and Law, 2010, name the drying time and the drying temperature as proportional factors of the thermal damage of the product [15,16]. Additionally, the drying temperature was mentioned as the most important influencing factor on the product quality. 

Pulsed Electric Field (PEF) pre-treatment represents a promising technology to achieve energy- and time-saving drying processes with high quality end products in comparison to the conventional drying methods [4,17,18,19,20,21]. It is a promising emerging pre-treatment for enhancing extraction or drying processes due to promoted mass transfer rates and therefore increased the yield or reduced drying times [4,22,23]. Recently, Ostermeier et al., 2018, examined the optimal PEF parameters for onion drying. Furthermore, it was shown that applying PEF as a pre-treatment can result in up to 30% drying time reduction for onions dried at constant temperatures. It is noteworthy that, as a consequence of this, the additional energy input for the PEF treatment (0.2−20 kJ/kg) is comparatively low considering the energy input for the actual drying process (4–6 MJ/kg detached water) [4]. Therefore, this processing step is of great interest for the drying industry to save energy, reduce costs, increase throughput, and reach sustainability goals.

However, most studies dealing with the influence of PEF on the drying kinetics and quality of fruits and vegetables focus on constant drying temperatures [24,25,26,27,28,29,30]. Very little is known about the drying kinetics and quality aspects of onions dried at drying profiles considering different drying steps as applied in industry. 

Therefore, this study deals with characterizing and optimizing the drying process for PEF-treated and untreated onions at temperatures and drying profiles that are relevant for industrial application. The different drying behavior is determined at various drying temperatures comparing treated and untreated onion samples. Various process and quality parameters will be examined applying individual drying profiles that reflect industrial conditions. The impact of a PEF pre-treatment on the convective drying process and quality of onions will be evaluated.

## 2. Material and Methods

### 2.1. Onion Samples

Onions (*Allium cepa*) from Spain, procured from a local whole-sale merchant (Osnabrück, Germany), with an average diameter of 81.5 ± 3.5 mm were used. The onions were stored at 10 °C and allowed to acclimatize to ambient temperature prior to the trials (max. 2 weeks). The average initial moisture of all examined onion samples was 90.2% ± 1.2%. For each series of experiments, four untreated and corresponding PEF-treated onions were peeled, and the first two fleshy layers were removed to eliminate outer damage from the harvesting, transportation, and storage of the onions. Afterwards, the third layer of each was stamped out to discs (*n* = 100) of 16 mm diameter and 3 ± 0.3 mm thickness. Two discs of each of the four onions were used for the determination of the initial moisture content. The measurement of all moisture contents was performed with a moisture scale (VPB-10, Allscales Europe, Wijk en Aalburg, Netherlands). 

### 2.2. Pulsed Electric Fields (PEF) Treatment

The PEF-Cellcrack II batch system (Elea Vertriebs-und Vermarktungsgesellschaft mbH, Germany) was used for the PEF treatments. It was equipped with a generator creating high-voltage exponential decay, monopolar pulses. These pulses were applied to the product in a batch treatment chamber consisting of two parallel stainless-steel electrodes at a distance of 300 mm. The pulse shape was monitored using an oscilloscope (Pico Technology, United Kingdom). The treatment cell was filled with tap water (initial conductivity 600 µS/cm), which served as a conductor between the electrodes. The total mass inside the cell was set to a value of 6.3 kg. As a consequence of sample quantity variation due to natural fluctuation and the use of whole onions, the mass inside the cell was maintained constant by adjusting the amount of water added. In each experiment, peeled onions (±1.2 kg) were processed. For reasons of comparability and for considering a possible water uptake of the onions whilst being in the treatment water, the untreated onions were put into a container filled with tap water for the same amount of time. Dabbing all onions gently with a paper towel, the adhesive water was removed from the surface afterwards.

According to the previous study from Ostermeier et al., 2018, a number of 56 pulses was applied to the product, resulting in a sufficient treatment with an electric field strength of 1 kV/cm and a specific energy input of 4 kJ/kg. The pulses were applied with a frequency of 2 Hz, the pulse width was 40 µs (measured at 37% of the peak voltage). Each PEF treatment was applied to two onions at the same time. The specific energy was calculated according to [17,31].

Applying PEF mainly addresses the onion tissue. Prior studies have shown that the cell size and distribution of the onion cells can impact the PEF-induced damage within the onion [32]. Furthermore, in multilayered plant material, the spatial location of the same cell type is an important factor that influences the PEF-induced cell disruption [33]. Higher PEF treatment intensities applied in this study are especially less affected by cell size and cell distribution, allowing a sufficient cell disintegration for the followed drying processes [17].

### 2.3. Convective Drying Process

All discs of one series of experiment were mixed and spread to drying containers (4 PEF; 4 untreated), the structure of which ensured a uniform drying and weighing of the samples. The drying containers consisted of a frame covered with a thin plastic net and a mesh size of 3 mm allowing air to flow through it. The drying of the untreated and PEF-treated samples on top of the containers was done simultaneously and repeated three times. The total mass load per drying cycle was approximately 70 g. Each slice had an initial mass of 0.9 ± 0.3 g. The clearance between the slices varied from 3 to 5 mm. No slices lay on top of each other.

As the drying industry uses different drying zones with various temperatures and residence times to keep the drying time short but product quality high, the first task of the study was to investigate the drying kinetics of the onions [10]. As PEF pre-treatment alters the product characteristic and drying behavior in terms of an increase in cell disintegration as well as a facilitated moisture release and diffusion, it is important to adapt the drying profile to achieve all the process and quality benefits [17]. Initially, the first and second drying phase were determined by defining the breakpoint (BP) after which the product temperature starts to increase. Therefore, the drying containers and samples were weighed with the precision scale (Kern 440-49A, Kern & Sohn GmbH, Germany, accuracy 0.1 g). Then, all samples were dried in a preheated drying oven (FP 240, Binder GmbH, Germany). The residual moisture *M_r_* of the sample was calculated according to industrial requirements of a certain maximum level of residual moisture left in dried goods [34]. The target moisture of all samples was set to ≤7%, as this moisture level is common in the onion industry. The values of *M_r_* vary between 100% and 0%, where 100% would be a sample, which consists completely out of moisture and 0% would be a total dry product. The moisture ratio (*MR*) was calculated for these experiments as well. It is known as the standard method to describe drying kinetics. However, in this particular case, only the *Mr* values are shown in this study as it is the standard unit used in the industry and it allows a second method to detect the different drying stages. This method to determine the BP includes the calculation of the slope of the residual moisture [10,34]. The slope represents the percentage drop of *Mr* in minutes between one measuring point followed by the next measurement point. If the slope reaches the highest intensity, the BP is reached.

For the BP determination, the drying process was performed at an air velocity of 0.2 m/s and constant temperatures of 85, 75, and 65 °C. Those slightly higher temperatures in comparison to the first paper (45, 60, 75 °C) were chosen as it turned out that the industry is especially interested in short drying times achieved by high temperatures. Hence, this study also investigated if higher temperatures in combination with a PEF treatment allow for time reduction without quality losses.

The surface temperature of the investigated onion discs was measured by means of an infrared thermometer (IR 100, PCE Holding GmbH) each 10 min. Additionally, to obtain the drying curve, all containers were weighed in uniform intervals of 10 min.

The delaypoint (DP), caused by a reduction in the drying temperature at the BP was evaluated as well. In order to determine the DP and a suitable drying temperature profile, the drying behavior of the onion samples was investigated by applying different drying temperatures. As the industry reduces the temperature till the end of the drying process to avoid quality losses, the samples were dried at 85 °C for the first drying phase and at the BP the drying temperature was reduced to 75, 65, or 55 °C. The drying process, analysis, and calculation were the same as one for determining the BP. 

### 2.4. Analysis of Quality Parameters

#### 2.4.1. Diffusion Coefficient

The calculation of the diffusion coefficient is performed in order to obtain information on the impact of the PEF treatment and resulting influence on the mass transfer behavior of the onion tissue during dehydration Equation (1). Furthermore, the same approach for the ion diffusion was conducted to investigate the rehydration behavior of the dried samples in water.

The effective water and ions diffusion coefficient *D_w_* and *D_i_* were computed with the Table Curve 2D software according to simplified Fick’s second law for an infinite disk of thickness *h* [35,36]:(1)$$y_{w,i}=y_{w,i}^{\infty }\left(1-\overset{n_{m}}{\underset{n=0}{\sum }}{\left(2n+1\right)}^{-2}exp\left[-{\left(2n+1\right)}^{2}t/\tau_{w,i}\right]/\overset{n_{m}}{\underset{n=1}{\sum }}{\left(2n+1\right)}^{-2}\right)$$
where $y_{w,i}^{\infty }$ is the maximum (saturation) level of the variable in the limit of *t*→∞, *τ_w,i_* = *(h/**π*)^2^/*D_w,i_* is the characteristic diffusion time, *h* is the thickness of the slice (m), and *D_w,i_* is the effective diffusion coefficients (m^2^/s). The subscripts *w* and *i* refer to the time dependencies of water and ions diffusion (s), respectively. Here, *n_m_* is the maximum number of terms in the series. For satisfactory fitting, the value *n_m_* = 10 was used. 

It should be noted that *D_w_* was separately calculated for the sections between the start of drying until the BP was reached and from the BP until end of drying. *D_i_* was calculated for the whole rehydration time.

#### 2.4.2. Colour and Blisters

To characterize the colour change and the browning, 10 g of grinded samples (Moulinette CH580, Kenwood, for 10 s) were measured using a spectrophotometer (CM-5, Konica Minolta). The field of view was set to 10° for a standard observer. The used method of the applied spectrophotometer is based on the CIEL*a*b*-system. The colour values, expressed by the L*a*b*-values as well as the colour differences Δ*L**, Δ*a**, and Δ*b** between PEF pre-treated and untreated samples were analyzed. By means of Equation (2), the colour difference Δ*E* between the PEF pre-treated and untreated samples was calculated. [37].
(2)$$\Delta E=\sqrt{{\left(\Delta L^{*}\right)}^{2}+{\left(\Delta a^{*}\right)}^{2}+{\left(\Delta b^{*}\right)}^{2}}$$

In order to determine the amount of onion discs with blisters, the number of samples showing blisters out of 100 PEF-treated and untreated onion discs after drying were counted manually and expressed as a percentage. Blisters bigger than 1 mm by 1 mm were considered.

#### 2.4.3. Shrinkage

The volumetric change of the samples during the drying process is expressed by the degree of shrinkage *S* and was calculated by Equation (3) [38].
(3)$$S=\frac{V_{i}-V_{f}}{V_{i}}$$
where *V_i_* is the initial volume of the fresh sample (mL) and *V_f_* the final volume of the dried sample (mL).

The applied determination method is based on the volume displacement by the sample. As the displacement media quartz sand was filled into a measuring cylinder up to a defined mark. After emptying the cylinder, the sample, respectively, 130 onion discs for each repetition, was placed in the measuring cylinder. In order to prevent distortion by formed cavities, amounts of the samples and the quartz sand were added to the measuring cylinder alternately up to the defined mark. The volume of the remained quartz sand was assumed to be equivalent to the volume displaced by the sample.

#### 2.4.4. Pyruvic Acid

The pyruvic acid content of the samples was determined according to the method of Schwimmer and Weston, 1961 [39] with modifications recommended from Anthon and Barrett, 2003 [40]. Furthermore, the simplified assumption that there is exclusively mass transport in form of water during the drying process was considered. A total of 10 g of the dried onion samples was grinded (Moulinette CH580, Kenwood) for 10 s. After adding 10 mL of deionized water, the mixture was homogenized by use of a mortar and pestle. The solid and liquid phase were separated with a filter paper after 10 min wait time at room temperature. The clear supernatant was diluted with deionized water in the ratio 1:50. Afterwards, further preparation of the samples was performed as described by [39,40]. The absorbance of the developing colour was measured with a spectrophotometer at 515 nm wavelength (GeneQuant 1300, Biochrom Inc., Cambridge, UK). To calculate the actual pyruvic acid content, a calibration curve with standards of different concentrations was prepared according to [41]. Note that, for the conversion of the units, the density of the onions was assumed to be the density of water. The concentration of pyruvic acid in the sample was than expressed as μmol/g dry matter. 

#### 2.4.5. Rehydration

A total of 5 g of the sample was gently mixed in 400 mL of deionized water. The weight of the sample was measured by sieving the samples out of the water and carefully shaking off the adhesive water before transferring them into the weighing pan. The total rehydration time was 330 min.

The ratio between the lost moisture during the dehydration process and the resumed moisture while rehydration was occurring can be expressed by the rehydration coefficient *RC* (%). *RC* was calculated corresponding to Equation (4) [42].
(4)$$RC=\frac{M_{r}-M_{d}}{M_{f}-M_{d}}\times 100\%$$
where *M_r_* is the mass of the rehydrated sample (g), *M_d_* is the mass of the dried samples and *M_f_* the mass of the fresh samples (g).

To estimate the degree of extraction of ionic components, the electric conductivity of the onion water suspension was monitored (Cond 3310, Xylem Analytics).

The initial electric conductivity of the deionized water before adding the onion samples was 0.7 µS/cm. The kinetics of the extraction were controlled measuring the electric conductivity of the water periodically during the rehydration time.

### 2.5. Statistics

All experiments and measurements of characteristics were repeated using, at least, four replicates. The statistical data analysis was performed by means of RStudio software version 0.99.893. The data were analyzed by a two-way variance analysis (ANOVA). The significance level was 95%. The error bars presented on the figures correspond to the standard deviations.

## 3. Results and Discussion

### 3.1. Constant Drying Temperature and Breakpoint (BP) Determination

For determining the different convective drying stages and the breakpoint (BP) for PEF-treated and untreated onion samples, constant drying temperatures between 85 and 65 °C were applied. 

Figure 1A–C shows *M_r_* and the surface temperature over the drying time at the constant drying temperatures. Due to an analogous behavior of the samples dried at different constant temperatures, only the slope curves for 85 °C are shown in Figure 1D. 

It can be seen that after a time of around 10 min the samples reached a constant temperature, which is typical for the initial drying stage [10,34]. However, it is noticeable that the temperature of the PEF-treated samples reached after 10 min is lower compared to the untreated samples. An explanation for this might be the facilitated water leakage to the surface as a result of the cell disintegration of the tissue by PEF [43,44]. A higher amount of surface water will result in more evaporation and therefore cooling of the surface. The stagnation in temperature was present in the range of 70 to 130 min for the PEF-treated sample and 90 to 120 min for the untreated sample after the temperature increased for the remaining drying time. This temperature increase indicates the transfer from the first into the second drying stage and is marked as a breakpoint for all drying temperatures in Figure 1a–c. Moreover, the final temperature for the PEF-treated samples (85 °C ≙ 51.1°C; 75 °C ≙ 60.4 °C; 65 °C ≙ 46.5 °C) was always lower than for the untreated samples (85 °C ≙ 73.1 °C; 75 °C ≙ 64.8 °C, 65 °C ≙ 56.0 °C). This might be caused by the PEF-enhanced surface moisture leakage resulting in greater evaporation cooling and the general shorter drying times necessary for the PEF-treated samples to reach a residual moisture *Mr* ≤ 7%.

The greatest drying time decrease after PEF treatment from 324 (untreated) to 178 min (PEF) (−45.0%) is present at a drying temperature of 65 °C. The drying time of the samples at a temperature of 75 °C decreases from 218 (untreated) to 151 min (PEF), which correlates to a percentage decrease of 30.7%. At 85 °C a drying time reduction of 27.6% from 152 to 110 min is present applying PEF. This allows a gentle drying, with potentially fewer quality losses at 75 °C after applying PEF, having nearly the same drying time in comparison to an untreated sample dried at 85 °C.

The trend of less drying time reduction caused by PEF at higher drying temperatures is in good agreement with previously made studies [15,17,45,46]. On the contrary, longer drying times at 75 °C for the untreated samples in comparison to the PEF-treated sample resulted in a greater reduction, comparing this paper with the first paper out of this study [17]. In the previous study, the untreated samples dried at 75 °C reached the final moisture after 154 min (PEF 144 min), whereas in this study the samples finished after 218 min (PEF 151 min) of drying time. A possible explanation might be the difference of the raw material properties, although the same variety of onions was used. Since the studies were conducted at different times of the year, the growing, pre-harvest, and storage conditions might have influenced the initial moisture content, the cell characteristic (turgor), and the thickness of the fleshy onion cell layers. 

Another way to characterize the drying stages is illustrated in Figure 1D. It includes the calculation of the slope of the residual moisture exemplarily shown for the sample dried at 85 °C. The greatest value of the slope (−1.24 L/min) for the PEF-treated sample was reached after 70 min; however, the untreated sample reached a slope of −0.95%/min after 90 min. This implies a 23.4% higher slope of the PEF-treated sample at the BP. A higher slope means a faster reduction in residual moisture, which might explain a 11.9% lower residual moisture for the PEF-treated sample at the BP in comparison to the untreated sample. 

As seen in Figure 1, both methods (temperature increase and slope of *Mr*) resulted in similar times for the BP of PEF-treated and untreated onion discs dried at 85 °C. For the samples dried at 75 and 65 °C, the same behavior was observed (slope not shown), although the BP was reached later due to lower drying temperatures. It should be mentioned that the *Mr* at the BP for the PEF-treated sample (24.5% ± 1.6%) dried at 65 °C showed with 64.7% less moisture the biggest difference to the untreated equivalent (69.5% ± 2.2%). This also demonstrates a higher impact of the PEF-induced moisture leakage while drying at lower temperatures. This is in good correspondence with the previously reported data [15,17,45,46]. 

Note that, for further trials, the initial drying temperature was set to 85 °C as it showed the fastest drying behavior without quality changes until reaching the BP (initial quality analysis not shown). In addition to this, the industry prefers higher temperatures at the beginning of the drying process to reduce the microbial load of the product to be dried [47].

### 3.2. Drying Profile and Delaypoint Determination

At each BP for the PEF-treated and untreated samples, the drying temperature was reduced to 75, 65, and 55 °C and maintained for the remaining drying process until reaching a final *Mr* of ≤7% (Figure 2A–C). Due to analogous behavior, Figure 2D shows only the slope curves of the samples dried at 85/65 °C.

Until reaching the BP, the curves behave like dried at constant 85 °C. However, when the drying temperature is reduced at the BP, the surface temperature of the PEF-treated and untreated samples decreases. A greater difference between the first and second drying temperature results in a lower surface temperature drop. This can be observed comparing for example the PEF-treated sample dried at 85/55 °C with a temperature drop of 6.9 °C and the PEF-treated sample dried at 85/75 °C with a drop of only 1.5 °C. Furthermore, the lower temperature plateau is maintained longer for the untreated samples (e.g., 10 min longer for the untreated sample dried at 85/65 °C). The point that marks the end of this plateau is the DP after which *T* increases again.

Moreover, equal to the BP determination, the DP was determined by analysis of the slope from the *Mr* curve. In contrast to the results for the samples dried at constant temperatures, a second drop, representing the DP, is present for the curves of the PEF-treated and untreated sample. The PEF-treated sample, with −1.05%/min (BP) after 60 min and −0.85%/min (DP) after 90 min, shows a greater slope that is reached earlier in the drying process compared to the untreated sample, with −0.59%/min (BP) after 90 min and −0.62%/min (DP) after 120 min. It can be summarized that both methods (temperature increase and slope of *Mr*) resulted in similar times of the BP and DP for PEF-treated and untreated onion discs dried at the drying profile 85/65 °C. For the samples dried at 85/55 °C and 85/75 °C, the same behavior was observed (slope not shown). This proves that, for all drying experiments, two independent methods determined similar time points to define the drying profiles for further analysis.

The surface temperature of the PEF-treated samples dried at 85/65 and 85/75 °C increases faster till the end of the drying process. A possible explanation could be the 25.2% (85/65 °C) and 4.4% (85/75 °C) lower *Mr* content at the DP for the PEF-treated samples and with it an overall facilitated drying process. A faster drying rate can be explained by an enhanced mass transport due to the electroporation [4,48]. However, with 51.5%, the greatest difference of the *Mr* content is present between the untreated (50.3%) and PEF-treated (24.4%) samples dried at 85/55 °C. Considering the difference in moisture between the samples and the bend of the *Mr* curve at the BP, which is especially pronounced for the untreated sample dried at 85/55 °C, the considerably low drying temperature of 55 °C also has an influence on the surface temperature and drying rate. This shows that the external parameter influencing the drying rate (mainly the drying temperature) and the internal parameter influencing the drying rate (mainly the diffusion of the water to the surface of the sample) are not in an equilibrium state. Applying PEF increases the diffusion rate towards an equilibrium state. However, a higher temperature (65 and 75 °C) after the BP is necessary to increase the evaporation rate and reduce the bend even more (Figure 2B,C).

With a drying profile and lower temperatures at the end of the drying process, the final product temperature was lower for all samples. This is intended from the industry to keep the product quality high. The temperatures at the end of the drying process for the PEF-treated samples (85/75 °C ≙ 61.5 °C; 85/65 °C ≙ 52.1 °C; 85/55 °C ≙ 43.8 °C) are between 3.1 and 5.0 °C lower than for the untreated samples (85/75 °C ≙ 66.0 °C; 85/65 °C ≙ 55.5 °C; 85/55 °C ≙ 46.9 °C). With those final temperatures, all samples are within or even below the temperature range of 55-66 °C suggested by [10,49], allowing a gentle drying with low quality losses.

As already observed for the experiments with a constant drying temperature, PEF treatment also allowed a significant reduction in drying time for drying with a temperature profile. The greatest drying time decrease after PEF treatment from 324 to 191 min (–41%) is present at the mildest drying temperatures of 85/55 °C. The lowest reduction in drying time with –21% can be seen between the PEF (143 min) and untreated (182 min) sample dried at 85/75 °C. The drying process for the PEF-treated sample dried at 85/65 °C was 32% shorter than that for the untreated sample with final drying times of 145 and 214 min, respectively. It is visible that applying PEF as a pre-treatment allows to dry gentler with a drying profile of 85/65 °C and even 5% faster when comparing with the untreated sample dried at a constant 85 °C.

It should be noted that, with reduced drying temperature, the residual moisture became lower at the BP and DP for PEF-treated samples. Nevertheless, the untreated samples showed a contrary behavior. *Mr* at the DP increased with decreasing drying temperatures (Figure 3). As a result, the gap between the PEF-treated and untreated samples became more visible at lower temperatures. Comparing all three drying profiles (85/75 °C, 85/65 °C, and 85/55 °C), the difference of *Mr* between PEF pre-treated and untreated samples increased from 4.4% to 25.2% and to 51.5%, respectively. A reason for this might be that PEF treatment not only allows for a facilitated mass transfer in the first drying stage until reaching the BP/DP, but also improves the second drying stage. The second drying stage is characterized by the migration of moisture from the inner interstices of the sample to the outer surface [10,34]. This limiting factor can be reduced by the PEF-induced pores in the membranes and a facilitated mass transfer [50,51].

### 3.3. Diffusion Coefficient

The PEF-treated samples showed a higher *D_w_* in comparison to the untreated samples for all drying profiles (Figure 4). The *D_w_* values until reaching the BP (drying at 85 °C) are constant for the untreated and PEF-treated samples. This is in good correspondence with the experimental setup as the temperature was changed at the BP and changes of the *D_w_* are expected afterwards. The average of the *D_w_* for the PEF-treated samples (1.28 × 10^−10^ m^2^/s) is 61% higher in comparison to the average of the untreated samples (5.00 × 10^−11^ m^2^/s). The fast drop of the residual moisture for the PEF-treated samples (Figure 2) confirms this.

For the drying period after the BP, the *D_w_* increases. As expected, the highest values with 3.65 × 10^−10^ and 2.18 × 10^−10^ m^2^/s are reached during drying at 85/75 °C for the PEF-treated and untreated samples, respectively. 

A general higher *D_w_* is present for all samples in the second drying phase when the temperature was reduced after the BP. A possible explanation is the increasing product temperature until the end of the drying period for all samples. With an increase in the air heat supply and higher product temperature, the internal diffusion is also higher [52]. Until reaching the BP, the surface of the samples is saturated with moisture that hinders the product temperature to heat up, resulting in a lower internal diffusion.

The PEF-induced accelerated migration of the water inside the onion also results in nearly the same drying time and *D_w_* for the PEF-treated samples dried at 85/75 and 85/65 °C. In both cases, the PEF treatment maxes out the evaporation of the water, as there is, for the majority of the drying time, always water available on the surface of the sample.

The estimated values for *D_w_* are in correspondence with previously obtained values for the effective water diffusion coefficient [17,46,53]. 

However, the *D_w_* values with a range from 4.78 × 10^−11^ to 3.65 × 10^−10^ m^2^/s are slightly lower in comparison to the initial study with values from 3.72 × 10^−9^ to 1.80 × 10^−8^ m^2^/s [17]. Mota et al., 2010, reported estimations of the effective diffusion coefficient in the range of 3.65 × 10^−9^ m^2^/s and 9.53 × 10^−9^ m^2^/s. The deviation between the studies might be explained by the difference in the drying process. Both used constant drying temperatures, which might result in a higher diffusion, especially in the second drying stage. A drying process with the reduction in the drying temperature after the BP as applied in the industry might lower the diffusion rate, but results in quality benefits as observed in this study. Furthermore, Mota et al. 2010, used a higher air velocity which results in facilitated drying kinetics in general [54].

The increase in the drying temperature increases the rate of drying and the effective diffusivity of water as well. This is in accordance with the work from Lewicki, Witrowa-Rajchert, and Nowak, 1998 [55], who also reported *D_w_* values in the same order of magnitude as in this study. Lewicki, Witrowa-Rajchert, and Nowak, 1998, conducted trials with a drying temperature drop from 80 to 60 °C. However, they were resulting in the conclusion that the effective diffusivity increases with increasing temperature, but it is strongly dependent on water content (high water content ≙ high *D_w_*, low water content ≙ low *D_w_)*. This could not be observed in this study, as the *D_w_* is higher for the drying period with low water content after the BP until end of drying. A possible reason for this contradiction might be the assumption in the calculations of Lewicki et al., 1998 [56], that the change in volume of the sample is equal to the volume of evaporated water. For the calculations in this study a constant thickness without shrinkage consideration was used, as this appeared to be in closer correspondence to the obtained experimental results. Furthermore, this study investigates the effect of PEF treatment on onion drying, which facilitates the internal diffusivity in general and therefore results in such high *D_w_* values, even at a lower sample moisture content.

To sum up, a PEF pre-treatment of onions facilitates the drying process in general. This is valid for the first drying step, but also for the second drying step, especially when the temperature is reduced due to a gentler drying. This confirms the statement that adapting the drying conditions is necessary to convert the PEF-induced tissue disintegration into an increase in process efficiency and quality benefits. 

### 3.4. Quality Analysis

#### 3.4.1. Colour

The colour as an apparent quality parameter for the food industry was investigated. The differences in colour between the PEF-treated and untreated samples are presented in Table 1.

The PEF-treated samples dried at 85/75 and 85/55 °C were slightly darker and more yellowish compared to the untreated samples. In contrast to this was the PEF-treated sample dried at 85/65 °C. It showed with *∆L** = 0.36 and *∆b** = −1.38 a slightly lighter and less yellow colour. A trend of the influence of PEF treatment on dried onions could be demonstrated by calculating the colour differences between the PEF-treated and untreated samples. *∆E* with 1.47 for the PEF-treated sample dried at 85/65 °C was the lowest overall colour difference compared to the untreated sample. The PEF-treated samples dried at 85/75 and 85/55 °C showed much higher differences with *∆E* = 5.65 and 4.28, respectively. As the samples exhibited Δ*E* > 2, the colour difference was perceptible for an untrained observer at a glance [57]. However, the colour difference between the untreated and PEF-treated sample dried at 85/65 °C (Δ*E* = 1.47) is referring to DIN 53230 classified as very small and thus barely visible by unaided eye.

A possible explanation for the colour change might be the enzymatic browning. Ebermann and Elmadfa, 2014, name the onion as the richest food source in the flavonol quercetin. A mechanical disruption of the onion cells during the chopping results in a contact of the enzymes peroxidase and polyphenoloxidase under the influence of oxygen. As a result, melanin is formed, a coloured end-product of the enzymatic reaction [58,59]. It is noteworthy that the colour change for both PEF-treated and untreated samples was more pronounced for the exterior section of the samples. This observation also corroborates the theory of the enzymatically browning process following the information of Ebermann and Elmadfa, 2014, that the flavonol quercetin has its highest amounts in the outer sections. An explanation for the more pronounced enzymatically browning for the PEF pre-treated samples might be the electroporation of the cell membrane [50,51]. Thus, the phenols of the onions and the enzymes come in contact to a greater extent.

Besides the enzymatic browning, a non-enzymatic colour change of the samples due to the exposure to heat can occur as well. For heat-related colour changes, the Maillard reaction has to be considered, which requires the presence of reducing sugars and amino acids [60,61]. Previous studies showed an increased leakage of soluble solids after PEF treatment out of plant material [43,62,63,64,65]. An enhanced leakage of sugars to the surface of the onion sample may allow an enhanced Maillard reaction, resulting in colour change.

Additionally, Lewicki, Witrowa-Rajchert, Pomaranska-Lazuka, et al. 1998 [56], stated that the drying temperature has a strong influence on non-enzymic browning. However, there was no statistically significant impact of the drying temperature on the lightness of the samples, expressed by *L**. This complies with the results of Krokida and Maroulis 1998 [66], who investigated drying temperatures between 50 and 90 °C for fruits and vegetables.

An explanation for the colour change of the PEF-treated sample dried at 85/55 °C might be the time of heat expose, meaning the overall drying time, that might resulted in colour changes [15,55,67].

According to the results obtained in this study, it can be said that the drying profile of 85/65 °C provides the comparative best combination of drying temperature and time in terms of the colour change comparing untreated and PEF pre-treated samples.

#### 3.4.2. Blisters

Whilst the percentage amount of onion discs with blisters was ≤1% for the PEF-treated samples throughout all drying profiles, the blister building for the untreated samples was more pronounced. With higher drying temperatures, the amount of onion discs that showed blisters increased. Consequently, the comparative highest number of blisters with 73.5% ± 1.5% occurred applying the drying profile of 85/75 °C for the untreated sample (Figure 5). The outer appearance of a representative PEF pre-treated and untreated onion disc dried at 85/65 °C is illustrated in Figure 5.

A reason for the high number of blisters on the untreated samples could be vapor creation under the product surface that cannot release quick enough. With increasing drying temperatures, more vapor is formed and therefore more blisters are present on the samples dried at high temperatures. In contrast to this, the PEF pre-treatment resulted in pores allowing the vapor to be released prior to build up and blister formation.

Moreover, a hiss was noticeable when the untreated samples were taken out of the drying oven. This could have been the result of the collapsing blisters due to the temperature difference between the drying oven and the ambient temperature.

#### 3.4.3. Shrinkage

The food industry demands a preferably high-volume retention of dried products. Thus, the volume change of the samples as a consequence of the drying process was measured and is depicted in Figure 6. The different drying profiles showed no significant impact on the shrinkage. Whereas, PEF treatment resulted in a significant higher shrinkage for the samples dried at 85/75 and 85/65 °C comparing to the untreated sample dried at the same drying profile. PEF treatment resulted in a 6.1% higher *S* comparing the PEF-treated and untreated samples dried at 85/65 °C with 92.0 ± 3.0% and 86.7 ± 2.8%, respectively.

Lewicki, Witrowa-Rajchert, Pomaranska-Lazuka, et al. 1998 [56] stated that the shrinkage of onion slices is anisotropic during drying. The same observation could be made in the present study. Another possibility might be the facilitated drying rate not allowing enough time for the food matrix to strengthen its structure, which results in shrinkage of the samples.

However, there are contradictory statements about the shrinkage of fruits and vegetables dried after PEF treatment. Parniakov et al. 2016, and Wiktor et al. 2016 [29,68], mentioned less shrinkage after PEF treatment for apples and carrots that were convective or freeze dried. On the other hand, Angersbach and Knorr, 1997 [21], Shynkaryk et al. 2008 [30], and Wiktor et al. 2016 [29], observed a greater shrinkage, depending on the PEF treatment parameters, for potatoes, carrots, and red beetroots after drying.

#### 3.4.4. Pyruvic Acid

The pyruvic acid amount is commonly used as a measure of onion pungency. Previous studies showed the formation of pyruvic acid as a consequence of the mechanical disruption of the onion tissue [39,69]. Moreover, considering that PEF treatment has a positive impact on mass transfer processes, it was of interest to evaluate the concentration of pyruvic acid in the treated samples. Therefore, Figure 7 presents the concentration of pyruvic acid in untreated and PEF-treated samples dried at different drying profiles.

In previous scientific publications, pyruvic acid contents between approximately 40 and 60 μmol/g dry matter were stated using drying temperatures of 50 to 65 °C [67,69]. This corresponds to the values determined in this study ranging from approximately 45 to 63 μM/g dry matter.

The pyruvic acid content of all samples decreased with decreasing drying temperatures. There was a trend towards higher pyruvic acid contents of PEF-treated samples in comparison to the untreated samples within the same drying temperature group. However, a statistically significant higher value within each drying temperature group could only be noticed for the drying profile of 85/65 °C. At 85/65 °C, the pyruvic acid content of the PEF-treated samples was 14.5% higher compared to the untreated samples.

The drying temperature and time had a strong influence on the pyruvic acid content of the onion samples. Samples dried at low temperature and therefore for a longer time showed a reduction in pyruvic acid. The decrease in the pyruvic acid content with decreasing drying temperatures complies with the findings of Sahoo et al., 2015. The reduction in pyruvic acid for longer drying times can be explained by the volatile behavior of the Sulphur compounds involved in the pyruvic acid formation. The general higher pyruvic acid content for the PEF-treated samples might be explained by the PEF-induced electroporation bringing the enzyme alliinase present in the cell cytoplasm into contact with the flavour precursors, *S*-alk(en)yl-l-cysteine sulphoxides. This will produce stoichiometric amounts of pyruvic acid [69]. 

#### 3.4.5. Rehydration

In order to examine the volume regression of the dried onion samples in water, the rehydration coefficient (*RC*) was calculated. For the rehydration trials the samples dried at 85/65 °C were used, as they showed the optimal balance between processing benefits and quality aspects.

Figure 8A shows the *RC* for the untreated and PEF-treated samples over a rehydration time of 330 min until *RC* for both samples approaches to plateau. From the beginning of the rehydration, the PEF-treated sample absorbs more water, resulting in a higher *RC* over the whole rehydration time. The final value for the PEF-treated sample is with 66.8% nearly twice as high as for the untreated sample with 35.1%. 

On the other hand, Figure 8B shows the conductivity of the water onion suspension over the rehydration time. The conductivity increased over a relative long time of extraction (100 min) for both samples, reaching values of 529 and 428 µS/cm for the PEF-treated and untreated onions, respectively. However, the PEF-treated onions showed a general higher degree of extraction throughout the whole rehydration time. This is supported by the increase in *D_i_* from 5.18 × 10^−10^ m^2^/s to 8.32 × 10^−10^ m^2^/s for the PEF-treated sample, indicating a higher amount of ion leakage throughout the whole rehydration.

This evidently reflects the development of electrically induced damage of the onion tissue and extraction of ionic components through the diffusion processes [61,70]. 

These findings are in close agreement with previous studies dealing with rehydration and extraction of ionic compounds after PEF treatment of fruits and vegetables [53,61,62,70].

Such an improved behavior due to PEF treatment during the rehydration is appreciated by the industry as dried onions are typically used in food products such as soups or sauces to be rehydrated. In such food preparations, onions are often used as an ingredient to add flavor or pungency into it. These findings show another aspect of quality improvement due to PEF-induced cell disintegration prior to onion processing.

## 4. Conclusions

The impact of a PEF pre-treatment on the convective drying and quality of onions has been investigated. It was shown that applying a PEF pre-treatment of 1.07 and 4.0 kJ/kg to whole onions prior to drying resulted in a positive change in the drying kinetics. For drying at a constant temperature of 85 °C, the BP was achieved 20 min faster. Moreover, PEF treatment resulted in a 25% faster drying process. This allowed us to reduce the drying temperature for the second drying stage earlier, still resulting in an efficient, but gentler, drying process. The best results with PEF in comparison to the untreated sample in terms of quality and process efficiency were achieved by applying the 85/65 °C drying profile. To reach a moisture of ≤7%, a reduction in drying time of 32% was achieved by applying PEF. Moreover, the evaluation of the effective water diffusion coefficient revealed that the effective water diffusion could be increased from 5.39 × 10^−11^ to 1.25 × 10^−10^ m^2^/s and from 1.99 × 10^−10^ to 3.48 × 10^−10^ m^2^/s in the PEF-treated tissue for the first and second drying section, respectively.

Regarding the quality of the PEF pre-treated product dried at 85/65 °C, practically no blisters were present and the pyruvic acid content was 14.5% higher in comparison to the untreated sample. Furthermore, PEF pre-treatment resulted in an enhanced rehydration behavior represented by a 47% higher *RC*. Additionally, this work has proven that the application of temperature profiles, instead of applying constant temperatures only, in combination with the effect of the PEF pre-treatment, improves the drying kinetics and the final product quality. The obtained findings will contribute to the further optimization of the industrial production of dried onions and provide the basis for the implementation of a PEF pre-treatment.

## Figures and Tables

**Figure 1 foods-09-00512-f001:**
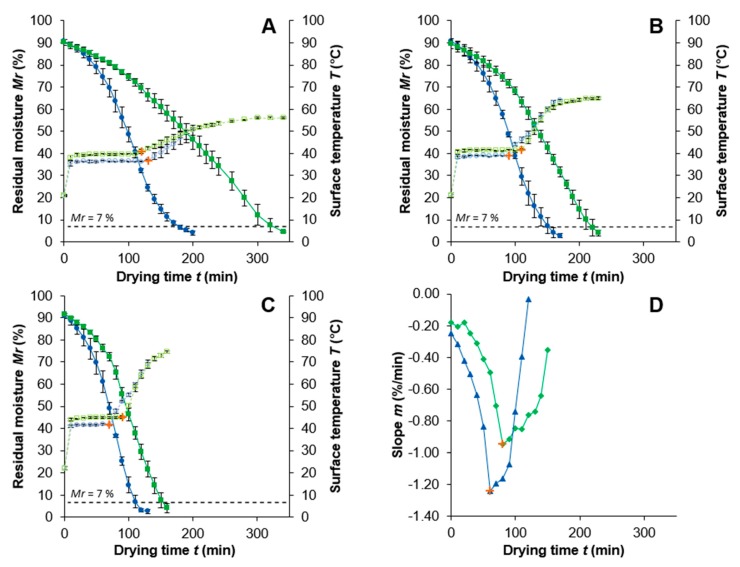
Surface temperature *T* (°C) (
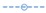
Pulsed Electric Field (PEF); 
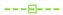
Untreated) and residual moisture *Mr* (%) (
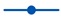
PEF; 
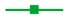
Untreated) for the drying time *t* (min) of onion samples dried at different constant drying temperatures (**A**: 65 °C; **B**: 75 °C; **C**: 85 °C). The PEF treatment was *W* = 4 kJ/kg, *E* = 1.07 kV/cm. (**D**): Slope *m* (%/min) of the residual moisture curves for the drying time *t* (min) dried at 85 °C (
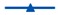
PEF; 
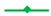
Untreated). 
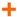
 represents the breakpoint (BP).

**Figure 2 foods-09-00512-f002:**
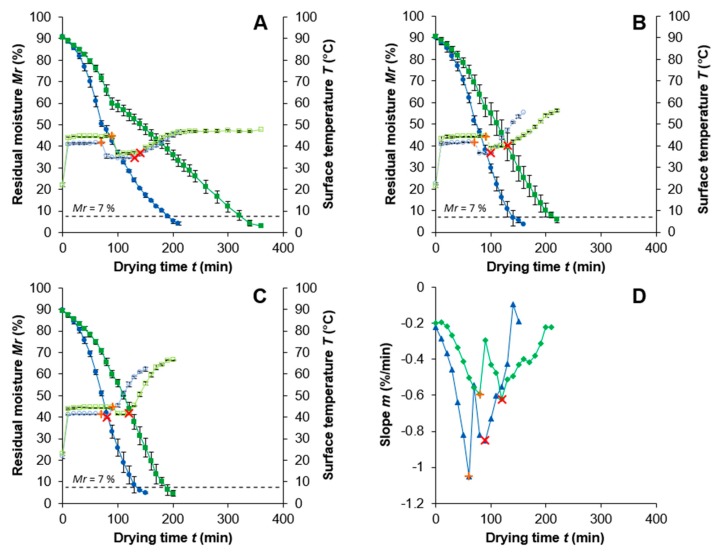
Surface temperature *T* (°C) (
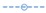
PEF; 
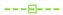
Untreated) and residual moisture *Mr* (%) (
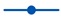
PEF; 
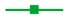
Untreated) for the drying time *t* (min) of onion samples dried at 85 °C until reaching the 
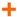
 breakpoint (BP) where the temperature was reduced (**A**: 55 °C; **B**: 65 °C; **C**: 75 °C). The PEF treatment was *W* = 4 kJ/kg, *E* = 1.07 kV/cm. (**D**): Slope *m* (%/min) of the residual moisture curves for the drying time *t* (min) dried at 85 °C and 65 °C (
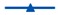
PEF; 
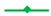
Untreated) 
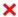
represents the delaypoint (DP).

**Figure 3 foods-09-00512-f003:**
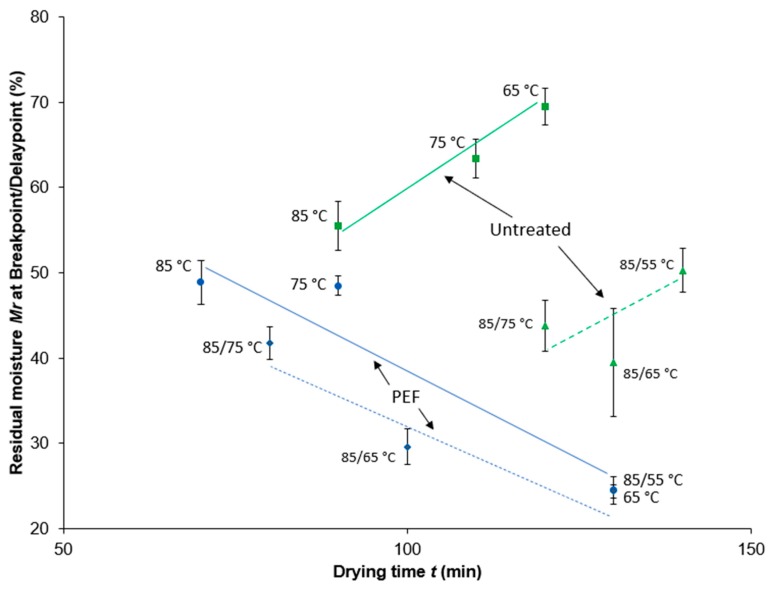
Residual moisture *Mr* (%) and drying time *t* (min) at the BP and delaypoint (DP) of untreated and PEF pre-treated (*W* = 4 kJ/kg, *E* = 1.07 kV/cm) onion samples dried at various drying temperatures (PEF ●; untreated ■) and drying profiles (PEF ◆; untreated ▲) for the necessary drying time *t* (min) to reach an *Mr* ≤ 7%. The lines are included for the guidance of eye.

**Figure 4 foods-09-00512-f004:**
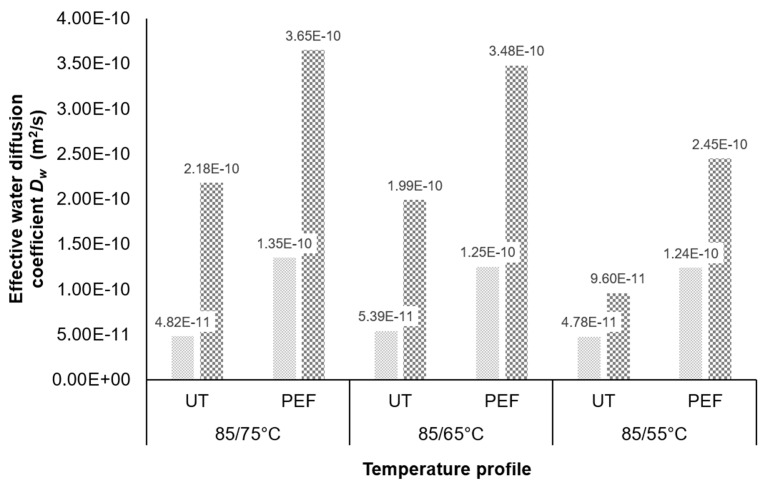
Effective water diffusion coefficient *D_w_* (m^2^/s) for the untreated (UT) and PEF pre-treated (*W* = 4 kJ/kg, *E* = 1.07 kV/cm) onion samples at different drying profiles. 
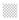
 represents the *D_w_* for 85 °C until reaching the BP, and 
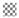
 represents the *D_w_* after the BP until end of drying applying a reduced temperature (75–55 °C).

**Figure 5 foods-09-00512-f005:**
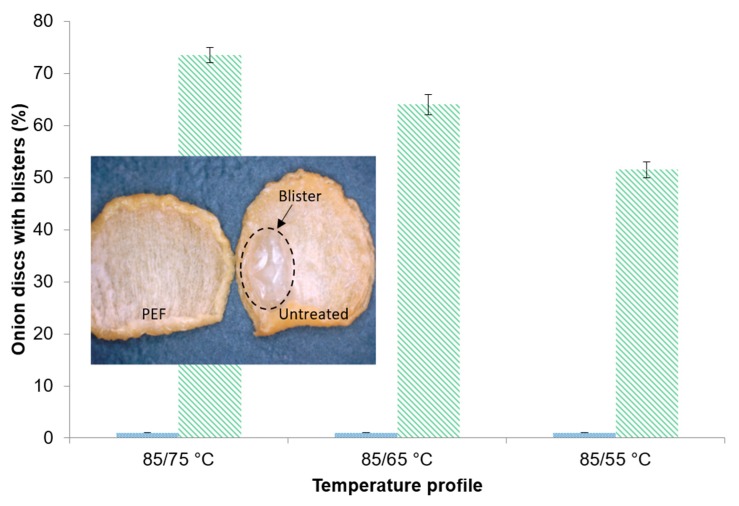
Onion discs with blisters (%) after drying from 
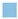
 PEF pre-treated and 
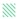
 untreated (*W* = 4 kJ/kg, *E* = 1.07 kV/cm) onion samples dried at different drying profiles until reaching an *Mr* of ≤7%. Exemplary footage of blister formation of an untreated sample in comparison to a PEF-pretreated sample.

**Figure 6 foods-09-00512-f006:**
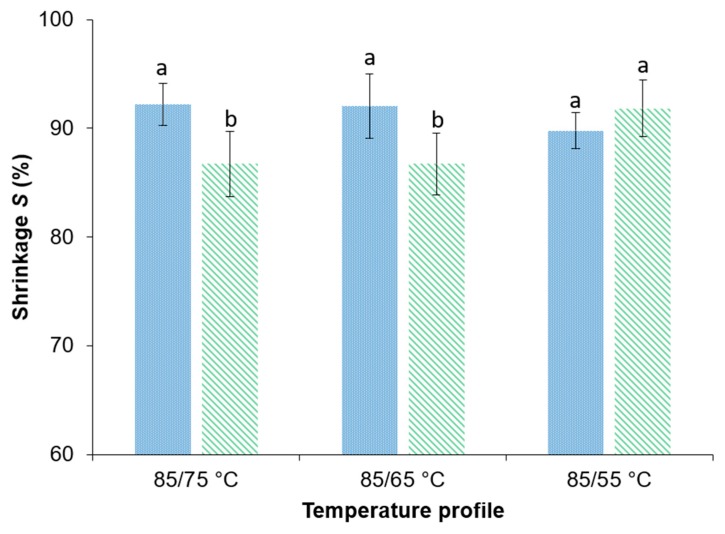
Shrinkage (%) of 
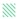
 untreated and 
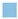
 PEF pre-treated (*W* = 4 kJ/kg, *E* = 1.07 kV/cm) onion samples with a residual moisture content of ≤7% after convective drying at different drying profiles. ^a-b^ For each bar, means followed by the same letter are not significantly different (*p* > 0.05). Statistical testing was performed for each temperature profile.

**Figure 7 foods-09-00512-f007:**
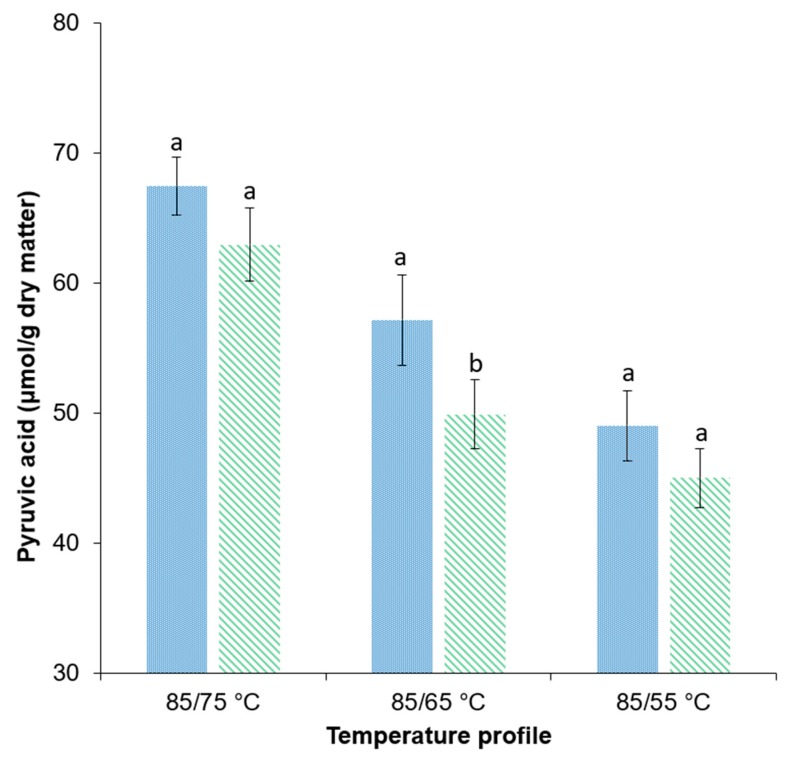
Pyruvic acid content (μmol/g dry matter) of 
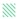
 untreated and 
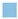
 PEF-pretreated (*W* = 4 kJ/kg, *E* = 1.07 kV/cm) onion samples with a residual moisture content of ≤7% after convective drying at different drying profiles. ^a-b^ For each bar, means followed by the same letter are not significantly different (*p* > 0.05). Statistical testing was performed for each temperature profile.

**Figure 8 foods-09-00512-f008:**
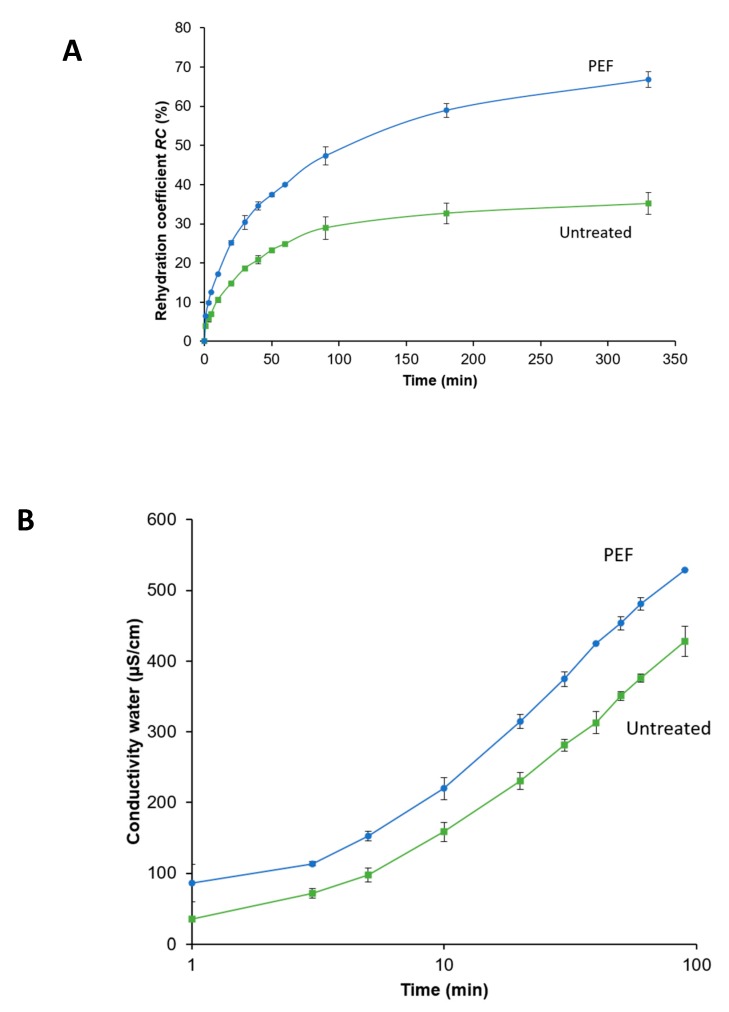
Rehydration coefficient (*RC*) (%) (**A**) and the conductivity of the water (µS/cm) as an indicator of the extraction of ionic compounds (**B**) of untreated and PEF pre-treated (*W* = 4 kJ/kg, *E* = 1.07 kV/cm) onion samples over the rehydration time (min). The samples were dried with the 85/65 °C drying profile.

**Table 1 foods-09-00512-t001:** Colour differences Δ*L**, Δ*a***,* and Δ*b** and the entire colour difference ∆*E* comparing the untreated with the PEF pre-treated (*W* = 4 kJ/kg, *E* = 1.07 kV/cm) onion samples at a residual moisture content of ≤7% after convective drying at different drying profiles.

Drying Profile (°C)	Δ*L** (–)	Δ*a** (–)	Δ*b** (–)	Δ*E* (–)
85/75 °C	−2.40	0.10	5.12	5.65
85/65 °C	0.36	0.36	−1.38	1.47
85/55 °C	−2.20	3.23	1.74	4.28

The colour values, expressed by the L*a*b*-values as well as the colour differences Δ*L**, Δ*a**, and Δ*b** between PEF pre-treated and untreated samples were analyzed, Δ*E*: the colour difference between the PEF pre-treated and untreated samples was calculated.

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
