# Peer review of "Applicability of Pulsed Electric Field (PEF) Pre-Treatment for a Convective Two-Step Drying Process"

_foods, 2020, doi:10.3390/foods9040512_

Round 1

Reviewer 1 Report

L 44 What about considerations related to the suitability of different onion cultivars for drying? What about the effects of seasonality etc?

L 55 Some explanations about the expected effects of PEF pre-treatment on the biological material, onion tissue, have to be included with appropriate references.

L 72 How was the initial moisture content determined? 

L97 The description is confusing. The text describes treating "peeled onions" and later, in 2.3, the convective drying describes discs.Please revise the M&M to clearly describe the experimental design and protocol

2.3 What was the min. max. and av. mass of the discs prior to drying? What was the among--discs difference in thickness prior to drying? 

L 105 What was the clearance between adjacent discs during drying?

L 109 Again, please, in what way the PEF treatment affected the onion tissue?

L 150 Why were the discs ground prior to color analysis? Using ground onion mass assumes no differences among the PSD of the samples. Based on what could this assumption been made?

L 162. Please provide a detailed description of the methodology that was used for the volume determination. Also, here and for all other analyses throughout the manuscript please provide the level of replication and the level of duplication (nXN).

L 255 Indeed, a weakness of the manuscript is the fact that only a very limited number of specimens (onions) were investigated. The potential effects of the inherent variability of onion properties, the potential effects of seasonality, cultivars, etc have neither discussed nor assessed. One could expect that when a significant deviation from previously reported trends or data becomes evident, especially when only a speculative explanation for the latter can be provided, more experiments with a larger population of specimens would be carried out. The authors are requested to address this aspect in light of the only speculative nature of many of the explanations that are provided.

L 270 A ref to substantiate the statement is required.

L 328 To what extent the reported differences in the final temperature that was obtained with the different treatments can actually lead to a significant influence on quality? Does a difference of less than 2 C carry a significant effect on quality? 

Throughout the discussion, the author assumed an even temperature distribution among the drying discs. Based on what was this assumption made?

 In general, the explanations about the potential effect of the investigated PEF pre-treatment on the drying properties are speculative and unsubstantiated. The authors are requested to provide better and substantiated explanations for their results.

Reviewer 2 Report

This article is interesting from scientific point of view. The PEF is new and effective methods applied in food technology.
The Authors should delete formula (1) or (2). This is this same.
Figures 4,5, 6, 7 are too big. The y axis don`t must started from "0".
For example Fig. 7: y skale can started from 30. Additionally the area of this figures are to big.
Above comments do not diminish the scientific value of this paper.

Reviewer 3 Report

The manuscript presents the effect of pulsed electric field (PEF) pre-treatment on convective two-step drying process of onion. Reviewed manuscript is well written and properly edited. Received results are supported by comprehensive discussion with other researchers.

Please explain why the onion was grinded before measuring the colour, the equipment gives the possibility of measuring onion discs?

Line 191-205: Diffusion coefficient should be presented as a point 2.4.1. It’s presented and discussion as a first (not the end) in the Results and Discussion section.

Line 225: please give unit i.e. 85 °C not 85 C.

Line 240: …from 324 to 178 min…please add (write in the text) with value is for untreated sample and PEF sample.

Line 267-268: Values in brackets, how were these values determined?? Unclear.

Line 399: …[53]… give full names of Authors of cited literature not number.

Line 428:….[55]… give full names of Authors of cited literature not number.

Line 526:….[67]… give full names of Authors of cited literature not number.

Line 576-580: Authors full names please replace on their initials.

Round 2

Reviewer 1 Report

The explanations that you have provided in responding to comments of the reviewer:

L44

L 55

L255

Should be integrated (with ref) into the revised manuscript. The fact that some of the requested information has been published in one way or the other in previous papers can substantiate the requested explanations. The results of the current manuscript have to be explained in light of what has been published. Highlighting the potential effect of the pre-treatment without providing acceptable and substantiated (by ref) explanations cannot be accepted. Please revise  
